# Analysis of Extreme Precipitation Events in the Mountainous Region of Rio de Janeiro, Brazil

**Maria del Carmen Sanz Lopez** [1,*], **Jorge Luiz Diaz Pinaya** [2], **Augusto José Pereira Filho** [1] , **Fe-lipe Vemado** [1] **and Fábio Augusto Gomes Vieira Reis** [3]

1   Instituto de Astronômia, Geofísica e Ciências Atmosféricas da Universidade de São Paulo, Departamento de Ciências Atmosférica, Sao Paulo 05508-090, Brazil; augusto.pereira@iag.usp.br (A.J.P.F.); fvfelp@gmail.com (F.-l.V.)
2   Polytechnic School, Universidade de São Paulo, Piracicaba 13418-900, Brazil; jorge.pinaya.usp@gmail.com
3   Instituto de Geociências e Ciências Exatas, Universidade Estadual Paulista, Rio Claro 13506-900, Brazil; fabio.reis@unesp.br
*   Correspondence: maria.d.c.s.lopez@gmail.com

**Abstract:** Extreme rainfall events cause diverse loss of life and economic losses. These disasters include flooding, landslides, and erosion. For these intense rainfall events, one can statistically estimate the time when a given rainfall volume will occur. Initially, this work estimated rainfall volumes for the mountainous region of Rio de Janeiro, and the frequency with which rainfall events occur. For this, we analyzed daily precipitation data using the ANOBES method and the Gumbel statistical distribution to estimate return times. Extreme prec'ipitation volumes of up to 240 mm per day were identified in some locations, with 100 years or more return periods. On 11 January 2011 precipitation volumes were high, but on 12 January they were extreme, similar to the 100-year return time data. The analysis method presented enables the determination of the return time of heavy rainfall, assisting in the prevention of its effects. Knowledge of the atmospheric configuration enables decision support. The atmospheric systems that combined to cause the event were local circulations (orographic and sea breeze) and large-scale systems (SACZ and frontal systems).

**Keywords:** rainfall; landslides; floods; extreme precipitation events





## 1. Introduction

Determining the intensity of precipitation is very important to control floods, land-slides, soil erosion, and other effects [1]. Rain is the most important type of precipitation in the Brazilian territory.

Disasters resulting from extreme rainfall events cause and have caused economic losses and of life. According to data provided by the WMO [2] in its Atlas of mortality and economic losses due to extreme rainfall events from 1970 to 2019, in this period, Brazil ranked second for major disasters in South America as a result of the flood of 11 and 12 January 2011 in the mountainous region of Rio de Janeiro which triggered landslides. This disaster also ranked 10th in the world for the last 111 years according to the UN [3].

When it is said that an extreme precipitation event occurs every return time, this implies that extreme precipitation events occur periodically and that there is a statistical probability that they will happen at a certain time. Thus, it is necessary to quantify these events and choose a statistical distribution that best represents them.

Local conditions, such as the operating atmospheric systems, influence the stability of the atmosphere; the morphological and isometric characteristics of the terrain also contribute to the effects caused by rainfall.

The objective of the study was to perform a comparative analysis of this flood event in relation to the extreme precipitation events in the region and to determine the return times

using the Gumbel statistical distribution [4]. The analysis of the flood event enabled estimation of the most probable period over which this volume of precipitation would be repeated.

Section 2 presents the procedure adopted for the analysis of the area. In Section 2.1 we describe the study area where the extreme precipitation event occurred, its geomorphological characteristics, and the atmospheric systems that act there. In Section 2.2 the method for estimating precipitation, ANOBES, is presented. In Section 2.3, preliminary statistics of satellite-based precipitation accumulation estimation for Rio de Janeiro State are presented. In Section 2.4, it is shown how the Gumbel statistical distribution is applied to precipitation return period analysis. Section 3 presents the results obtained. In Section 4, the results are discussed. Section 5 presents the conclusions of the study and suggests some potential applications.

## 2. Materials and Methods

The steps used for the proposed methodology are described below:

(a) Selection of the Serra Region of Rio de Janeiro as the study area, and determination of the atmospheric systems acting in this area as well as the geomorphological characteristics; (b) Collection of precipitation estimate data, by integrating satellite data and rain gauge data; (c) Analysis of extreme events by applying the Gumbel statistical distribution to obtain the return times.

The period of data collection for precipitation estimates was from 2000 to 2015. With the data obtained, we determined the maximum daily precipitation estimate for each year and, among these, the maximum daily precipitation estimate of the entire historical series (16 years). In addition to the maximum values of the historical series, it was necessary to calculate the mean and variance of the extreme values. The study area was divided into cells of 8 km by 8 km according to the data standard proposed by the climate prediction center morphing method (CMORPH).

The statistical distribution that best fitted the data was chosen, i.e., the Gumbel or double exponential statistical distribution. The scale ($\alpha$) and location ($\mu$) parameters and the estimated return time values of each cell in days were determined. The precipitation estimate values obtained were analyzed.

### 2.1. Study Area

The selected study area was the mountainous region of Rio de Janeiro, composed of the municipalities of Nova Friburgo, Petrópolis, Teresópolis, Bom Jardim, São José do Vale do Rio Preto, Sumidouro and Areal, in an estimated area of 2300 km², located from latitude 22° S to 22°48′ S and longitude 42° W to 43°30′ W, as shown in Figure 1.

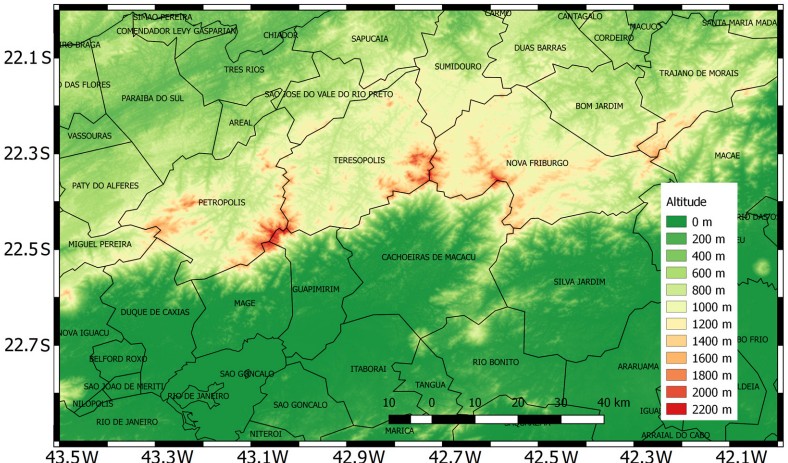

**Figure 1.** Extreme rainfall event study area in the municipalities of Nova Friburgo, Petrópolis, Teresópolis, Bom Jardim, São José do Vale do Rio Preto, Sumidouro, and Areal. Distance scales, latitude and longitude are indicated on the map. Color scale indicates altitude (m) from latitude 22° S to 22°48′ S and longitude 42° W to 43°30′ W.

When studying the geotechnical maps of susceptibility to gravitational mass movements and floods of these seven municipalities [5], it can be seen that the altitudes vary from 40 m to 2220 m, and in the central municipalities of the region (Petrópolis, Teresópolis, and Nova Friburgo) that this variation is more accentuated, as shown in Table 1; the slopes vary from 0° to more than 45°, whereas in Petrópolis, Teresópolis and Nova Friburgo slopes greater than 20° predominate.

**Table 1.** Geomorphological characteristics of the municipalities of the mountainous region of Rio de Janeiro State (Figure 1).

| Municipality | Altitude (m) | Slope above |
|---|---|---|
| Petrópolis | 40 to 2200 | 20° |
| Areal | 300 to 1020 | 20° |
| Teresópolis | 160 to 2200 | 25° |
| São José do Vale do Rio Preto | 400 to 1420 | 20° |
| Nova Friburgo | 160 to 2220 | 25° |
| Sumidouro | 260 to 1760 | 16° |
| Bom Jardim | 280 to 1620 | 25° |

Source: letter of susceptibility to gravitational movements of mass and flooding.

The soils with a greater slope have the predominant characteristics of being young, shallow, and poorly developed with abundant exposed rocky outcrops. The soils with slopes between 10° and 20° are moderately deep and well-developed. The soils with slope gradients <10° are well-developed and thick. In general, the slopes in the region are very steep. With very shallow soils, they are not very well able to withstand heavy rainfall. Deforestation also contributes greatly to the increased risk. With regard to flooding, the characteristics that can be highlighted are the occurrence of hydromorphic soils, on a flat terrain or with a slope below 2°; the terrain is an alluvial plain and the water level in the soil is at a shallow depth.

Three main factors have influenced the susceptibility of the mountainous region of Rio de Janeiro to landslides to date: the steepness of the slopes, the thickness of the soil, and the characteristics of the rocks. Considering the local atmospheric circulations, we have, due to the relief, orographic circulation, and, due to the proximity to the ocean, maritime circulation. In addition, as this region has a very pronounced relief considering the orography, there is an active atmospheric system. The orography can alter the distribution of precipitation, thus exercising a fundamental role in its distribution [6]. By analyzing the maps of precipitation distribution and hypsometric distribution, it is evident that the orography influences the spatial distribution of rainfall.

According to Mendonça and Danni-Oliveira [7] rains are classified according to their genesis, which results from the type of controlling process of the upward movements that generate clouds that can later precipitate. Orographic or relief rain is one category. One can define the process of orographic rain as involving the existence of a barrier to the free advection of air, forcing it to rise. Thus, the humid and warm air ascending near the slopes cool, saturating the vapor, and the ascension tends to produce rain.

Another circulation system that must be taken into consideration is the sea breeze. It is a diurnal thermal circulation system. It occurs as a result of the heating gradient between the land and the sea during the day, causing the land to heat up faster than the sea during this period. With this difference in air temperature between the continent and the sea, the wind blows in a sea to land direction and there is a tendency for cloud formation on the continent. At night the air circulation system is inverted causing the continent to cool down more than the sea [8].

Thinking about other larger-scale atmospheric systems that also have an influence on the highlands, there is the South Atlantic Convergence Zone (SACZ) [9]. It is recorded that an episode of SACZ occurred from 11–16 January 2011. The SACZ, acting together

with high sea surface temperatures (above average) off the coast of southeastern Brazil, contributed to an increase in air humidity and the occurrence of continuous rainfall. This episode of SACZ contributed to the disaster of 11–12 January 2011 [10].

Another larger-scale atmospheric system is the frontal system. The frontal system [11] occurs when, initially, there is a low-pressure cell over the ocean near the continent. It involves cyclonal movement and displacement leading to the formation of a frontal system that, over the mountainous region of Rio de Janeiro, is characterized as an occluded front. The occluded front occurs when the density of cold air is greater than that of warm air (from the tropical zone), forcing it to remain juxtaposed to the surface, opposing the upward movement imposed by the cyclonic circulation occurring in the center of low pressure. Thus, one can see the contact of the occluded front with the NW–SE instability line (SACZ). In this way, precipitation is associated with the contact of cold air with even colder air with the warm air above it forming cumulonimbus-type clouds generated by convective movements responsible for rain of greater intensity [11,12].

Atmospheric circulations acting together in this area can result in increased effects. Thus, the local atmospheric system can produce more intense extreme precipitation events.

### 2.2. Precipitation Estimation by the SOAS Method

Precipitation estimation quantification based on satellite measurements is an important product used worldwide for hydrometeorological applications, such as weather and climate monitoring and water resource management [13]. The CMORPH is a product that estimates precipitation with high spatiotemporal resolution using satellite measurements. This method enables the quantification of precipitation estimates with a temporal resolution of 30 min and spatial resolution in an 8 km × 8 km cell [14].

By integrating the data from the CMORPH and the data from the National Water Agency (ANA) surface rainfall network using the objective analysis and statistics (SOAS) method, the error variance is reduced, and more robust data is obtained. The data from the CMORPH and the rainfall network are integrated every 24 h, from 7 a.m. on one day to 7 a.m. on the previous day. Figure 2 shows a scheme that can guide the integration of CMORPH data with rain gauge data to obtain SOAS data [14,15].

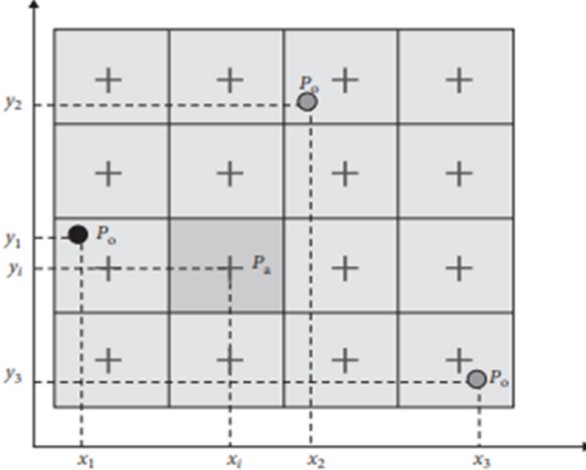

**Figure 2.** Scheme for the interpolation of ANA rain gauges with the CMORPH cell at the point where the erosion event (Pa) occurred. Reprinted/adapted with permission from Ref.: [14]. Copyright *2018, Adv. Meteorol.*

The calculation of the precipitation estimate is obtained for SOAS by applying Equation (1) below.

$$P_a(x_i, y_i) = P_b(x_i, y_i) + \sum_{k=1}^{K} W_{ik}[P_o(x_k, y_k) - P_b(x_k, y_k)] \sum_{k=1}^{K} W_k\left[\rho_{ik} + \varepsilon_i^2\right] = \rho_{io} \quad (1)$$

where,

$P_a (x_i, y_i)$ = precipitation analyzed (mm) at grid point $i$;
$P_b(x_i, y_i)$ = precipitation estimated with CMORPH (mm) at grid point $i$;
$P_o(x_k, y_k)$ = precipitation measured by rain gauge (mm) at point $k$;
$P_b(x_k, y_k)$ = precipitation estimated with CMOPRH (mm) at point $k$;
$W_{ik}$ = a posterior weights when there are more than two rain gauges;
$\rho_{ik}$ = background cross correlation between points i.e., $k$;
$\rho_{io}$ = background cross correlation between points i.e., $k$;
$\varepsilon_i^2$ = normalized error of measurements.
$K$ = number of rain gauges used
$(x_{i(k)}, y_{i(k)})$ = coordinates (km) of grid points $i$ and $k$.

### 2.3. Statistical Analysis of SOAS and CMORPH Precipitation Estimates

Rainfall estimates with CMORPH and those integrated with rain gauges using the ANOBES method were evaluated for the State of Rio de Janeiro. Figure 3 shows the monthly differences between the total monthly precipitation obtained with the ANOBES and CMOPRH methods for all months from January 2000 to December 2015. Note that the differences tend to be greater for larger monthly precipitation volumes. The states of Rio de Janeiro showed greater differences due to the greater influence of the Serra do Mar region. In general, the positive differences indicate that the CMOPRH tends to underestimate the accumulated precipitation.

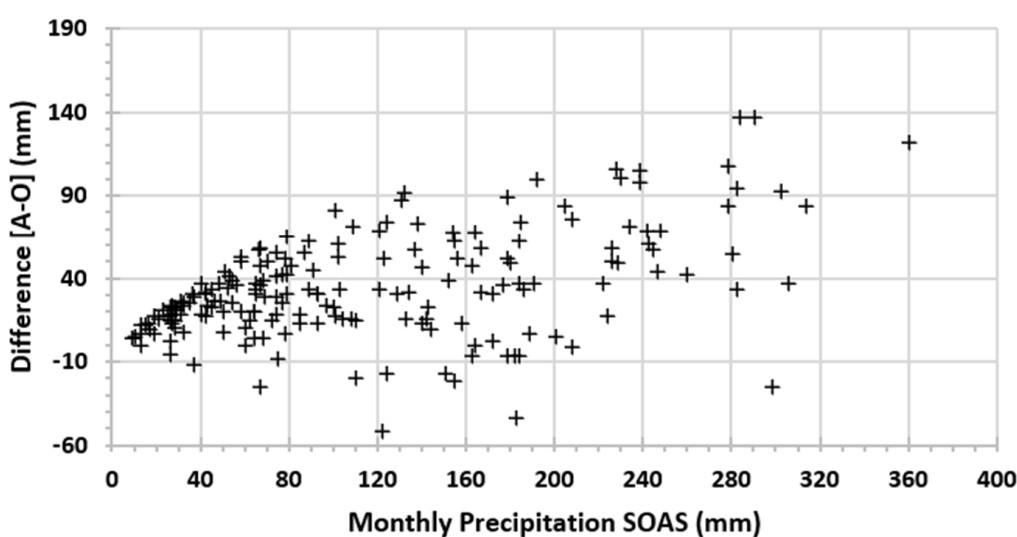

**Figure 3.** Monthly precipitation average difference (mm) between SOAS (A) and CMPORH (O) as a function of SOAS monthly precipitation average for all months between 2000 and 2015 for Rio de Janeiro State.

Figure 4 shows the monthly distribution of monthly precipitation totals between the years 2000 and 2015. The annual precipitation cycle is remarkable in the State of Rio de Janeiro with a rainy period in spring and summer and a less rainy period in autumn and winter. The period from 2000 to 2015 was marked by precipitation extremes. The monthly precipitation totals from month to month indicate great variability from year to year. Monthly rainfall in the State of Rio de Janeiro has greater interannual variability and even more rainy periods in the autumn and winter months.

Figure 5 shows the difference between the monthly precipitation totals obtained with ANOBES and the original CMORPH in the State of Rio de Janeiro between the years 2000 to 2015. The differences between ANOBES and CMORPH are more variable

throughout the year, but generally positive. It is suggested that the greater spectrum of precipitant systems from more stratiform (e.g., cold fronts) to more convective (e.g., CCM) results in underestimation of the second and overestimation of the first.

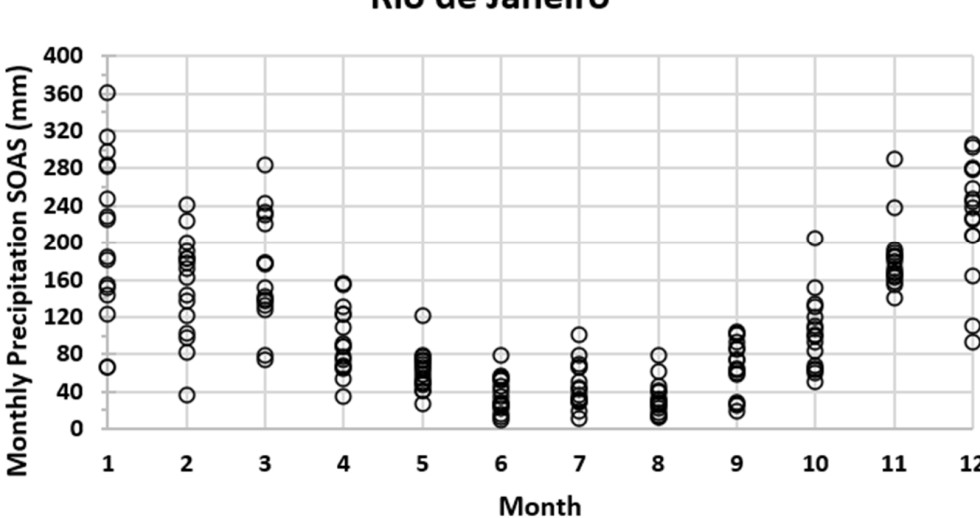

**Figure 4.** SOAS monthly areal precipitation averages between 2000 and 2015 for Rio de Janeiro State.

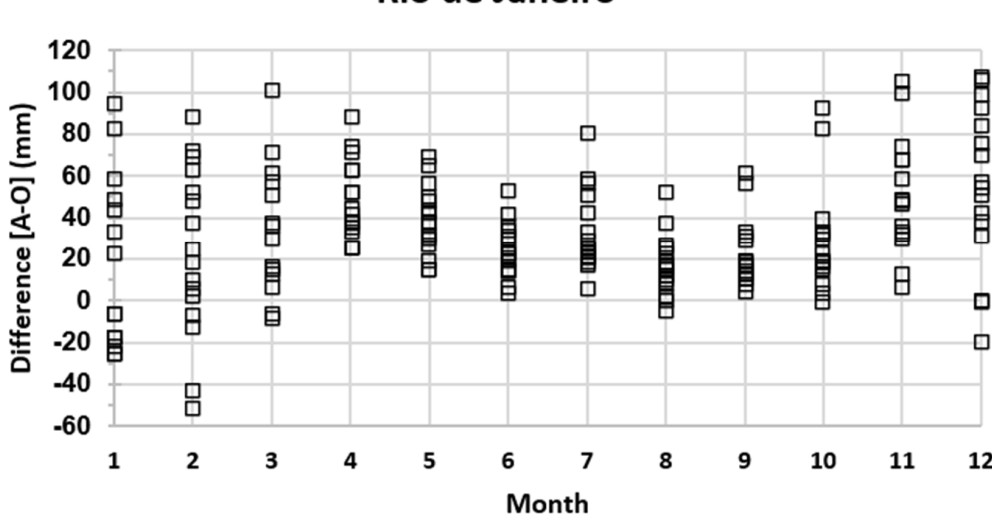

**Figure 5.** Monthly differences between the average monthly precipitation difference obtained with the SOAS method (A) and that estimated with CMORPH (O) for all months of the year between 2000 and 2015 for the State of Rio de Janeiro.

Figure 6 presents the spatial standard deviation of mean monthly precipitation as a function of mean monthly precipitation between 2000 and 2015 for the SOAS and CMORPH methods for the State of Rio de Janeiro. The dispersion diagrams were fitted by an exponential curve whose equations and coefficients of variation are indicated in the respective figures.

More significant changes are observed in both coefficients of the fitted curves and coefficients of variation with changes in means and standard deviations. Thus, the statistics of means and standard deviations change with the integration of rain gauges to the CMOPRH estimate, as there is a greater impact on this from local circulations in the coastal region, but also and especially due to the underestimation along the Serra do Mar. Mass movement events are more frequent and severe in this region and, therefore, there is an improvement in the characterization of these events using the SOAS database, including extreme statistics.

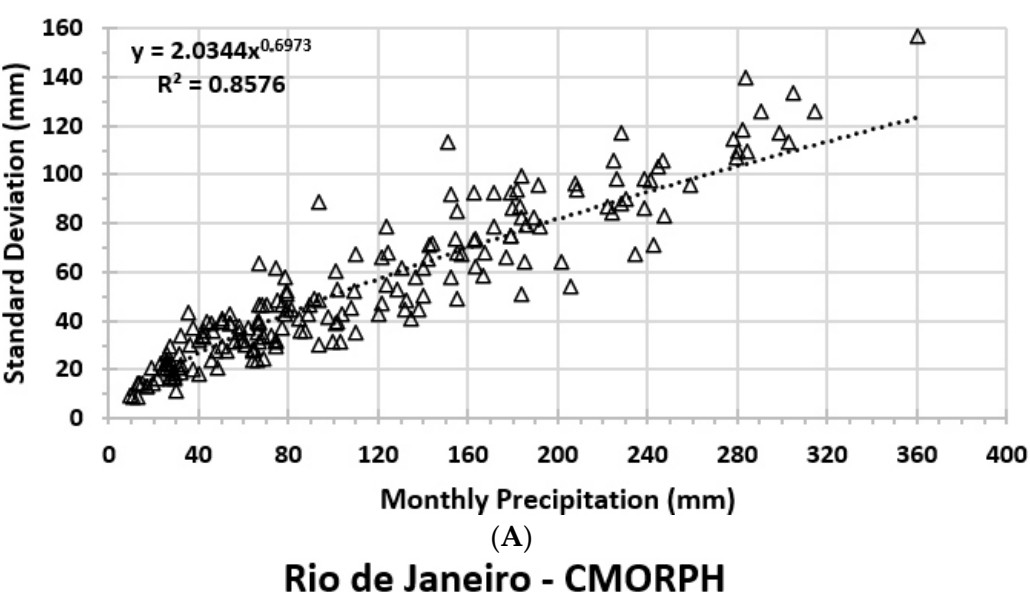

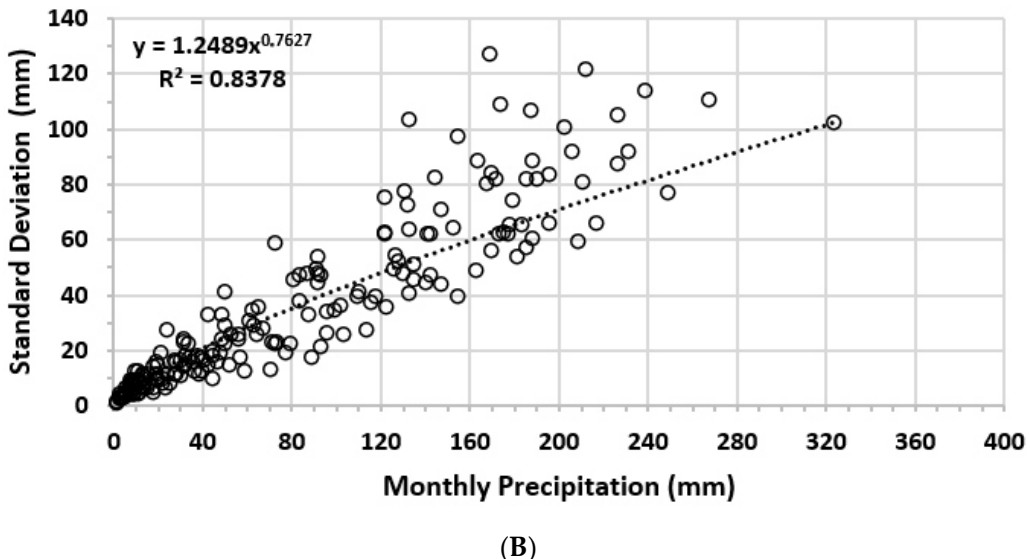

**Figure 6.** Scattering diagram between mean monthly precipitation and respective spatial standard deviation obtained with SOAS and the CMORPH method for all months of the year between 2000 and 2015 for the State of Rio de Janeiro. The exponential curve adjusted to the data, equation, and coefficient of variation adjusted by the method of least squares are indicated.

Figure 7 shows the scatterplots between the mean monthly precipitations obtained with SOAS and with CMORPH and between the respective monthly spatial standard deviations. In general, the highest monthly averages obtained with SOAS are greater than those obtained with CMORPH. These results reveal the positive impact on precipitation estimates through the integration of estimates obtained with the CMOPRH method and measurements from the state pluviometry network. Results of comparative analyzes between CMORPH and rain gauge networks suggest that the latter tend to underestimate rainfall totals.

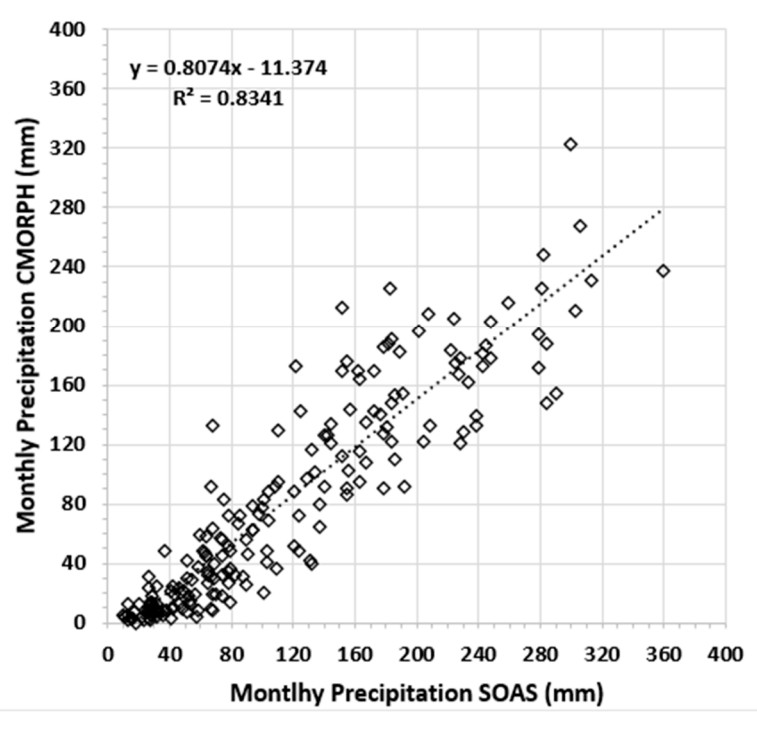

(**A**)

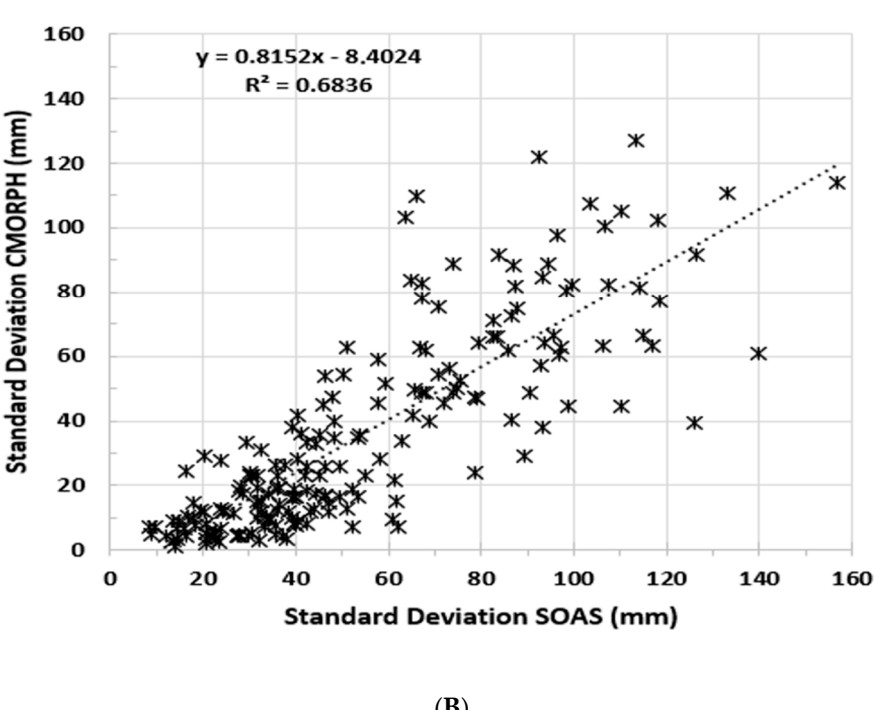

(**B**)

**Figure 7.** Scatter diagrams between mean monthly precipitation (**A**) and areal standard deviation of mean monthly precipitation (**B**) obtained with the CMORPH and SOAS method for all months of the year between 2000 and 2015 for the State of Rio de Janeiro. Linear adjustments, equations, and coefficients of variation adjusted by the method of least squares are indicated.

### 2.4. The Gumbel Statistical Distribution

The extreme value theory (EVT) originated in the work of Fisher and Tippett to describe the behavior of the maximum of identified independent random variables. Several applications have been successfully implemented in many fields, such as hydrology, climatology, engineering, economics, and finance. The extreme *e*-value distribution is, perhaps, the most widely used probability distribution in climate modeling, including for global warming, flood frequency analysis, rainfall modeling, etc. [4]. The return time (*Tr*) is obtained from the Gumbel statistical distribution from the calculation of

$$T_r = \frac{1}{(1 - P[Y \leq y])} \tag{2}$$

where,

$Tr$ = return time (years);
$P[Y \leq y]$ = probability of occurrence.
The cumulative probability function is given by:

$$P[Y \leq y] = e^{-e^{-\alpha(y-\mu)}} \exp(-\exp(-\alpha(y-\mu))) \tag{3}$$

where,

$Y$ = all rainfall values; (each of the maximum values);
$y$ = arbitrary rainfall limit (maximum value of maxima);
$\alpha$ and $\mu$ = distribution, scale, and location parameters [16].

### 3. Results

Initially, we extracted the estimated precipitation data from SOAS in the mountainous region of Rio de Janeiro, as shown in Figure 1. These data were placed in a table, with each cell showing its respective maximum estimated precipitation value for each year between 2000 and 2015. With these values (maximums), the maximum among the values of the series was determined, in addition to determining the mean and variance to use in the calculations of the parameters, $\alpha$ and $\mu$, and statistically determining the return time estimate.

Analyzing the daily data of precipitation estimates, it was verified that the rainfall that occurred on 11–12 January 2011 accounted for the maximum precipitation (mm) in many of the cells. Figure 8 shows the estimated precipitation accumulation (mm) in the study area for 11 and 12 January. It is noted that on 10 January the precipitation was very low or zero.

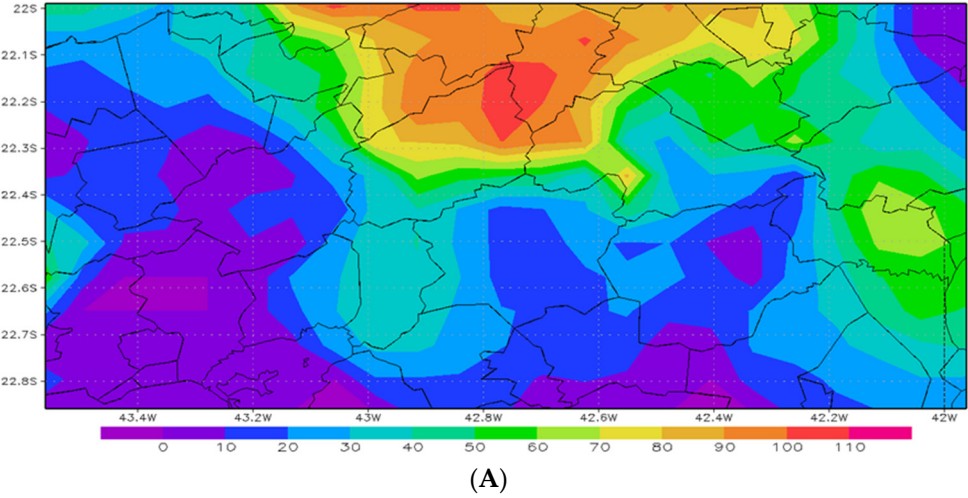

(A)

**Figure 8.** *Cont.*

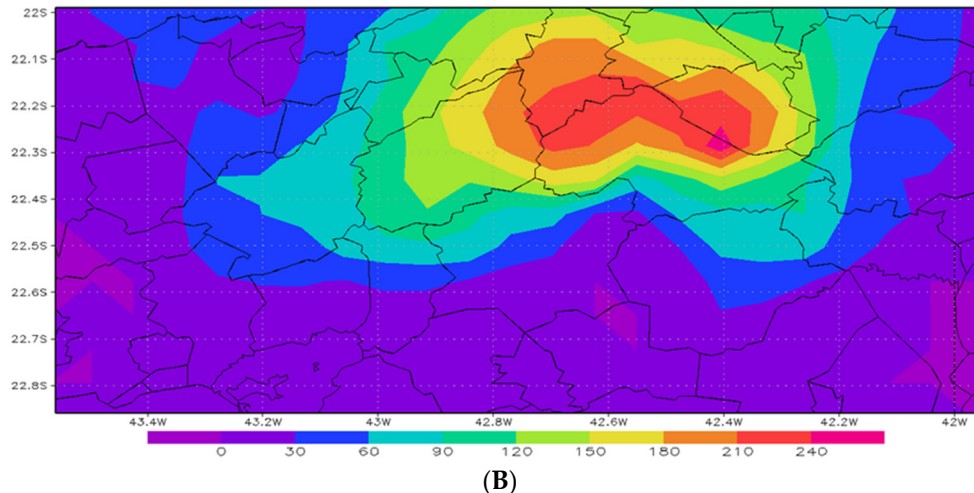

**(B)**

**Figure 8.** Estimation of daily precipitation in mm, with data from SOAS, at the coordinates of the mountainous region of Rio de Janeiro. (**A**) 11 January 2011, (**B**) 12 January 2011.

It can be observed that, on the 12th, the precipitation values reached 240 mm and in the same place on the 11th the precipitation values reached 100 mm; adding up the precipitation it can be seen that, in these two days, the precipitation reached 340 mm. Thus, these locations had excessive rainfall values, which accounts for what happened. Although the same color scale is not used between Figure 8A,B, a movement of the maximum precipitation (from day 11 to day 12) to the south can be observed. The municipalities most affected by the precipitation extreme are Sumidouro, Nova Friburgo, and Bom Jardim, with a daily value >240 mm between Bom Jardim and Nova Friburgo. In addition, there were landslide events occurring mainly in Nova Friburgo.

From analysis of the historical series of extreme events using the Gumbel statistical distribution, we obtained precipitation values for the return periods of 2, 5, 10, 25, 50, and 100 years, presented in Figure 9, and Tables 2 and 3. The maps presented relate the geographic coordinates with the return time values for the precipitation that occurred at the site.

**Table 2.** Example points. Columns 1 and 2 are the geographic coordinates; column 3 is the maximum value of precipitation of the historical series; column 4 calculates the value of the average of the years of the historical series; and column 5 is the value of the return time that the precipitated volume (mm) represents.

| Geographic Coordinates | | Series Value | | Calculated |
|---|---|---|---|---|
| **Latitude** | **Longitude** | **Maximum** | **Average** | ***Tr* (Years)** |
| −22.7955 | −42.9871 | 90.40 | 63.94 | 28.66 |
| −22.7955 | −42.9143 | 78.16 | 65.29 | 8.70 |
| −22.7955 | −42.8416 | 89.93 | 70.35 | 9.48 |
| −22.7955 | −42.7688 | 95.76 | 73.15 | 12.06 |
| −22.7955 | −42.6960 | 107.93 | 74.70 | 23.53 |
| −22.7227 | −41.9685 | 204.65 | 81.61 | 133.56 |
| −22.2862 | −42.6233 | 208.52 | 78.17 | 110.77 |
| −22.2862 | −42.5505 | 175.54 | 74.46 | 120.67 |
| −22.2862 | −42.4778 | 201.16 | 81.06 | 130.55 |
| −22.2862 | −42.4050 | 262.37 | 95.89 | 192.71 |
| −22.2862 | −42.3323 | 185.62 | 84.47 | 128.08 |
| −22.2862 | −42.2595 | 137.75 | 80.46 | 36.33 |
| −22.2862 | −42.1867 | 113.68 | 77.29 | 20.52 |

**Table 3.** Example points. Estimated precipitation values at different return times.

| Geographic Coordinates | | Volumes Corresponding to Return Times in Years (*Tr*) | | | | | |
|---|---|---|---|---|---|---|---|
| Latitude | Longitude | 10 | 20 | 25 | 50 | 75 | 100 |
| −22.7955 | −42.9871 | 79.98 | 86.88 | 89.06 | 95.80 | 99.97 | 102.49 |
| −22.7955 | −42.9143 | 79.40 | 85.47 | 87.40 | 93.33 | 96.99 | 99.22 |
| −22.7955 | −42.8416 | 90.61 | 99.32 | 102.09 | 110.61 | 115.86 | 119.06 |
| −22.7955 | −42.7688 | 93.38 | 102.08 | 104.84 | 113.35 | 118.60 | 121.79 |
| −22.7955 | −42.6960 | 96.43 | 105.77 | 108.74 | 117.87 | 123.51 | 126.93 |
| −22.7227 | −41.9685 | 129.34 | 149.87 | 156.38 | 176.44 | 188.83 | 196.36 |
| −22.2862 | −42.6233 | 131.04 | 153.78 | 160.99 | 183.22 | 196.94 | 205.27 |
| −22.2862 | −42.5505 | 114.62 | 131.89 | 137.37 | 154.25 | 164.68 | 171.01 |
| −22.2862 | −42.4778 | 127.90 | 148.05 | 154.44 | 174.13 | 186.28 | 193.67 |
| −22.2862 | −42.4050 | 155.40 | 181.00 | 189.12 | 214.13 | 229.57 | 238.96 |
| −22.2862 | −42.3323 | 124.09 | 141.14 | 146.55 | 163.20 | 173.49 | 179.74 |
| −22.2862 | −42.2595 | 112.40 | 126.14 | 130.49 | 143.92 | 152.21 | 157.25 |
| −22.2862 | −42.1867 | 102.46 | 113.28 | 116.72 | 127.29 | 133.83 | 137.79 |

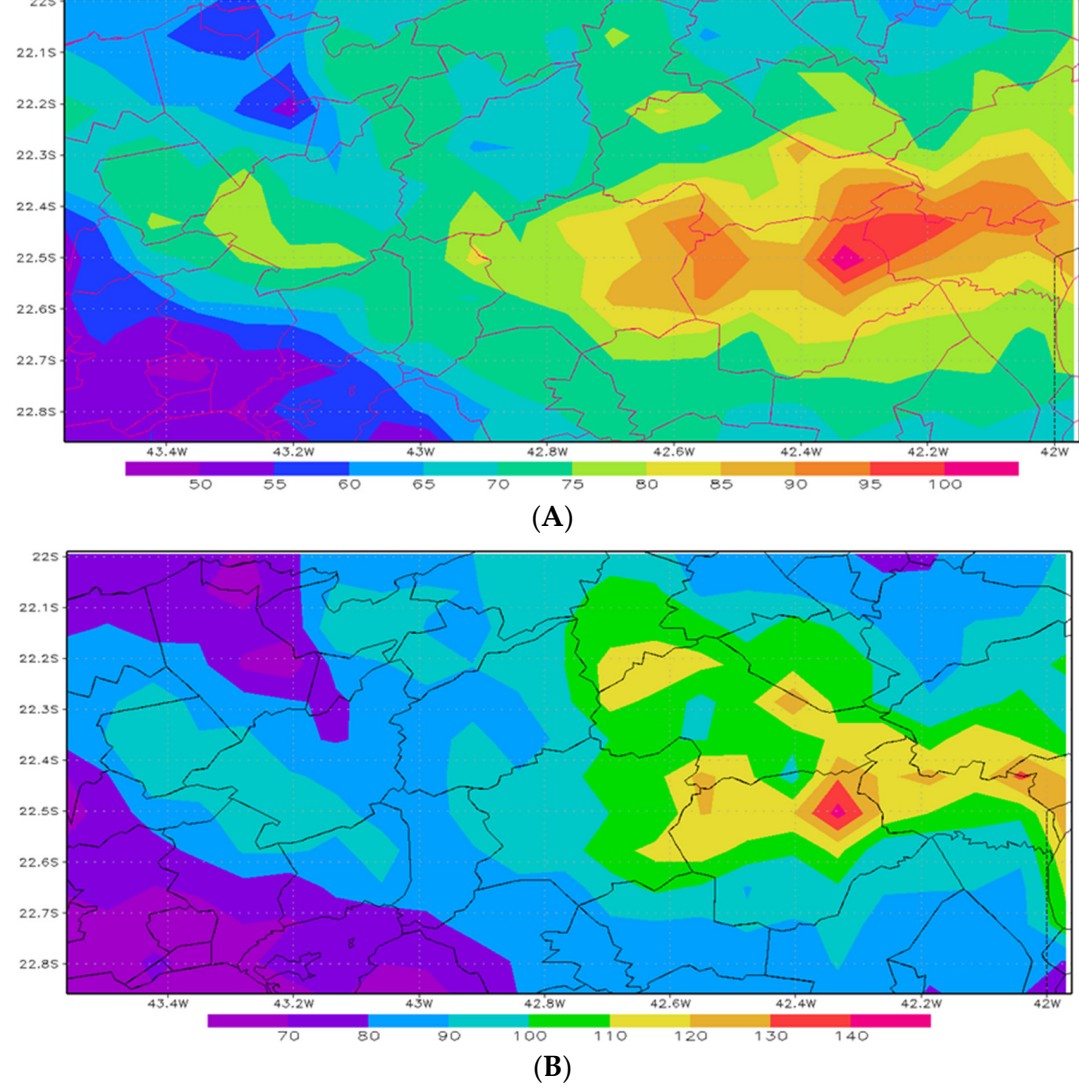

(**A**)

(**B**)

**Figure 9.** *Cont.*

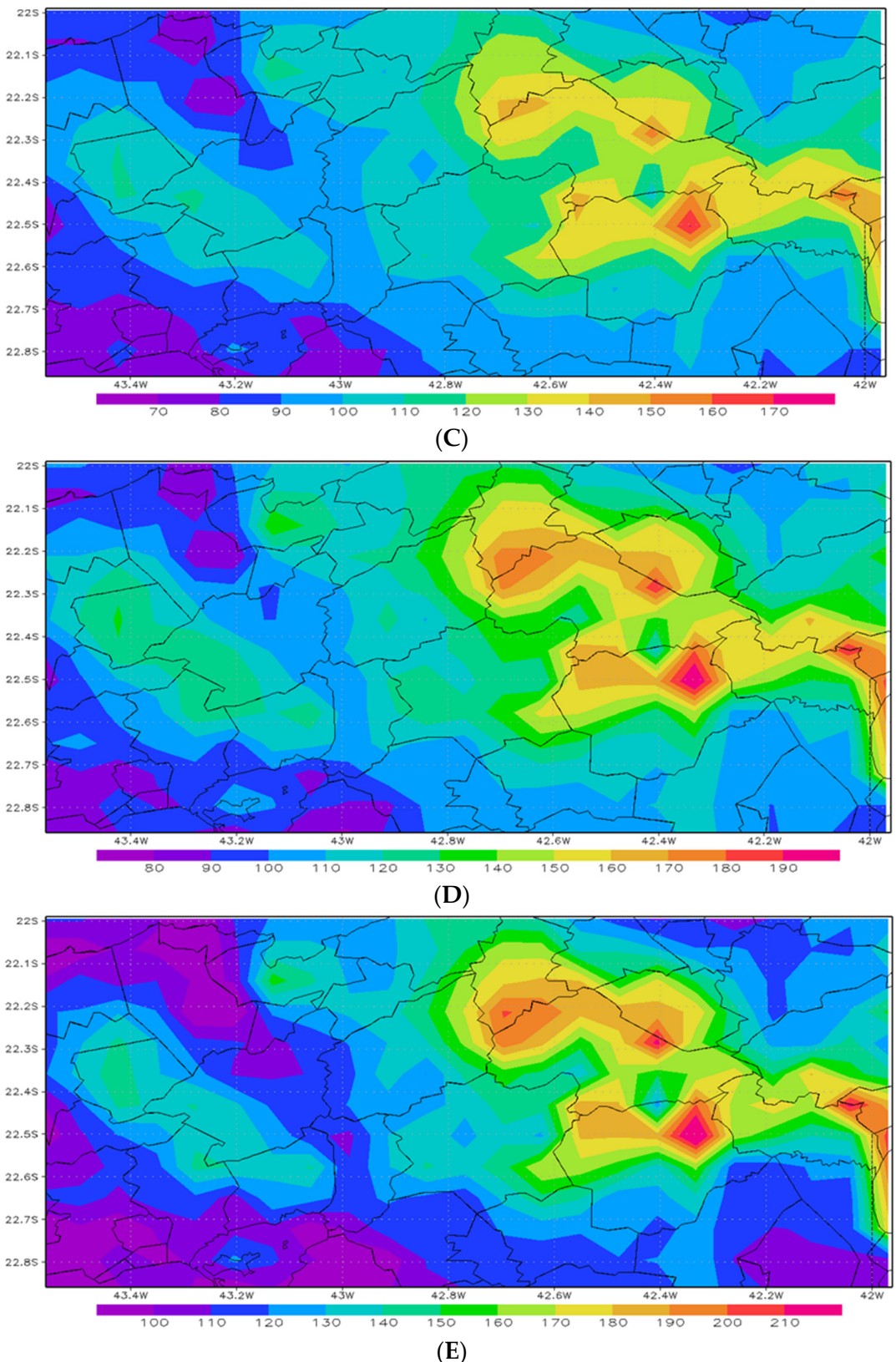

**Figure 9.** *Cont.*

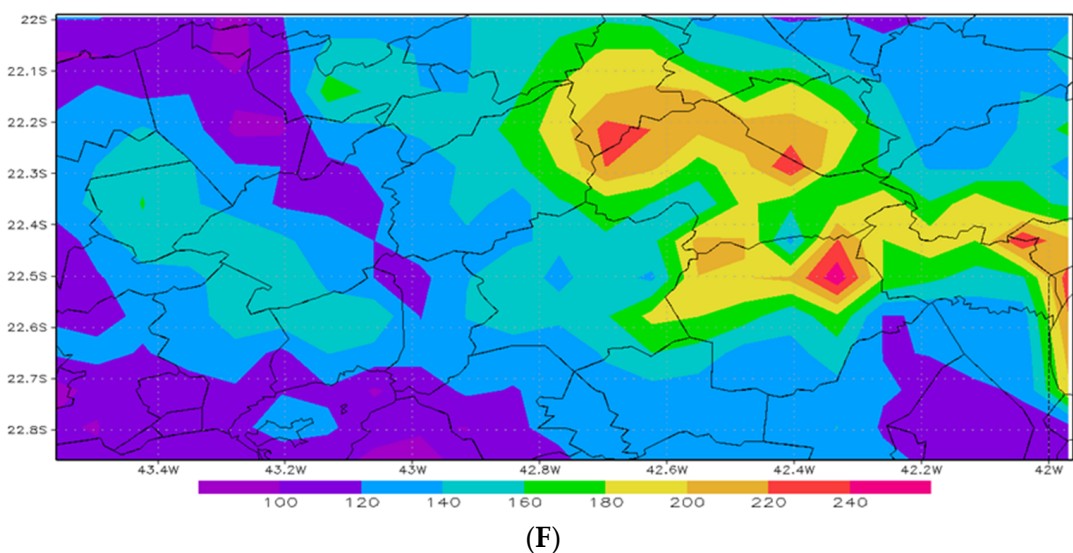

(F)

**Figure 9.** Spatial distribution of rainfall accumulation (mm) for return periods for 2 years (**A**), 5 years (**B**), 10 years (**C**), 25 years (**D**), 50 years (**E**) and 100 years (**F**), for the mountainous region of Rio de Janeiro (Figure 1).

It can be seen that the municipalities most affected by precipitation estimated in return periods of 2, 5, and 10 years are Sumidouro, Bom Jardim, and Nova Friburgo, with some municipalities further south where we do not have a very high altitude. These municipalities with lower elevation and inclination of their slopes (those further south) have a more developed soil structure, with greater thickness and older, increasing water infiltration into the soil and better drainage. If one compares Figures 8B and 9F, one can see a great similarity in the precipitation values on 12 January 2011 and the 100-year return time map.

## 4. Discussion

There is a large variation in precipitation throughout the area of the Serrana Region, with higher values in the north on the 11th, while, on the 12th, the highest precipitation values occur over the municipalities of Nova Friburgo, Bom Jardim, and Sumidouro. In these locations, the maximum accumulated precipitation reached 340 mm between the 11–12 January 2011. It can be seen in Figure 1 that this is a steep slope location, causing the precipitation that falls at the top of the mountain to gain speed dragging everything in its path (downhill). The soils that are on these large slopes are liquefied because they are shallow and cannot absorb as much water. The soils, or liquefied matter, move down the slopes changing the local landscape, and, therefore, the landscapes below on the slope. Depending on the force that the water gains, this matter will reach further away and change the landscapes in its path [17–19].

On 11 January 2011, the maximum precipitation corresponds to values of precipitation extremes around 5 years. Meanwhile, the 12 January precipitation values correspond to precipitation extremes with return periods around 100 years, as compared between Figures 8B and 9F.

The atmospheric circulations of orography and sea breeze types contribute to the occurrence of these extreme precipitation events by warming the atmosphere and making it more unstable. The existing soils at high altitudes (shallow and young, with rocky outcrops) and high slope gradients also contribute to landslides. The water infiltration in the soil quickly saturates it and, since there is bedrock under it, the soil is carried away by the superficial runoff. In addition to local and mesoscale circulations, large-scale systems also contribute to increasing rainfall amounts. The result of the acting systems (orographic circulation, sea breeze, SACZ, and frontal systems) caused floods and landslides in this event.

The state of Rio de Janeiro has several places that are prone to floods, landslides, and other disasters. Several studies have addressed the disaster in the mountainous region of

Rio de Janeiro. One of the studies analyzed the atmospheric system from the spatialization of precipitation data in a concomitant analysis of synoptic charts [11]. Another emphasized the disasters caused by a convergence of phenomena related to the interaction between society and the environment, such as the occupation of hills (urbanization). Other studies have analyzed only parts of the problem: the large-scale meteorological systems that combined to add their effects to local systems and, thus, maximized the disaster. In this study, we tried to consider all the factors, or most of them, that contributed to the occurrence of these landslides and floods, among other effects. Using the suggested method, an attempt was made to relate the volumes of precipitation to the effects of extreme precipitation. The return time of these volumes was estimated.

## 5. Conclusions

The geomorphological conditions of this mountainous region, associated with extreme precipitation events linked with local to large-scale atmospheric systems, are conducive to flooding and landslides. When there are extreme precipitation events, disasters occur [20]. However, analysis of the historical series of precipitation estimates and application of methods such as the Gumbel statistical distribution can enable estimation of their occurrence. The analysis method presented in this study enables the determination of the return time of intense rainfall, helping to prevent the effects of this type of event. It also provides precipitation estimates and return times. Analysis of the atmospheric configuration can provide warnings of the occurrence of such events facilitating prevention.

The high-impact precipitation event on 11 January 2011 was the worst in Rio de Janeiro State's history [21]. Different responses to the tragedy, such as forensic investigation [22], multiscale high precipitation thresholds [23], and geophysical susceptibility criteria [24] have been considered and proposed for the mountainous region of Rio de Janeiro State. The methodology used in the present work can be implemented to improve the above types of analyses, and to guide the planning of projects for the introduction of structural and non-structural measures for areas prone to mudslides, floods, debris flows, mass movements, etc., along the coast of Brazil and elsewhere.

**Author Contributions:** Conceptualization and methodology A.J.P.F.; software and validation, A.J.P.F., F.-l.V. and J.L.D.P.; formal analysis, M.d.C.S.L.; investigation, J.L.D.P.; resources and data curation, F.A.G.V.R.; writing—original draft preparation, M.d.C.S.L. and A.J.P.F.; writing—review and editing, M.d.C.S.L. and J.L.D.P.; visualization, F.-l.V. and J.L.D.P.; supervision, A.J.P.F.; project administration and funding acquisition, F.A.G.V.R. All authors have read and agreed to the published version of the manuscript.

**Funding:** This research was funded by Petrobras, Grant 2014/438-9.

**Data Availability Statement:** The datasets are available upon request to the first author.

**Acknowledgments:** This research was developed with datasets provided to the Petrobras Project (Grant 2014/438-9). The third and fifth authors were sponsored by Conselho Nacional de Desenvolvimento Científico e Tecnologico—(CNPq) (Grants 302349/2017-6 and 316574/2021-0, respectively).

**Conflicts of Interest:** The authors declare no conflict of interest.

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
