# Peer review of "Analysis of Extreme Precipitation Events in the Mountainous Region of Rio de Janeiro, Brazil"

_climate, doi:10.3390/cli11030073_

Round 1

Reviewer 1 Report

This manuscript attempts to analyze extreme precipitation events in the Mountainous Region of Rio de Janeiro, Brazil, using data for 16 years (2000 to 2015). However, the manuscript is not written properly, so it is not suitable for publication in a journal, among the main points are as follows:

1. Introduction is very poor. The research gap is unclear. The author should discuss research related to extreme precipitation events in the Mountainous Region of Rio de Janeiro, Brazil, and elaborated the research gaps.

2. The method is also inadequate. Study area, data and statistical tests used, should be written adequately. Data quality control should be adequately described.

3. Results and discussion chapters are also inadequate. The discussion is very minimal, predominately related to the rain events January 11 and 12, 2011. It is better for the author to focus only on the physical processes of the rain events January 11 and 12, 2011, not to discuss extreme rainfall in the long term. If this is done, the author must completely change the manuscript.

Author Response

A spelling review was carried out in the English language. The changes indicated below are made in the text.

Page 3 - line 94 - Table 1: completed and updated for English

Page 5 – 185 lines – changed the equation

Page 9 - line 276 is OK

Page 10 - line 281 included (B)

Page 11 – lines 357 and 358: added text

Page 14 – line 405 – landslide events and their scars were not the objects of this article

Page 14 – lines 441 to 453 – added text

Reviewer 2 Report

The manuscript titled “Analysis of extreme precipitation events in the mountainous region of
Rio de Janeiro, Brazil” shows very good results. The study has good potential. The methodology and datasets are quite acceptable. However, some following issues should be modified:

·        Is there any novelty found in the methodology of this study?

·        Abstract should be a summary of the study. It needs numerical representation.

·        Introduction section is too short. Many new articles should be discussed in this section. It must be rewritten.

·        A flowchart of methodology is required.

·        In the discussion section, the manuscript simply needs a comparison with the recently published articles. There are many recent similar research articles available on some global cities. Kindly consider these articles.

·        Conclusion section should be more analytical. Each objective should have a response.

Author Response

(The authors gave the same response as above.)

Reviewer 3 Report

A more detailed explanation regarding the use of data should be made.

Validation methods of extracted data and applied methods should be explained.

The writing language of the text should be corrected in terms of grammar and...

Explanation of the text should be done by providing various references.

The figures should be specified more precisely in terms of the location of the study area.

Author Response

(The authors gave the same response as above.)

Reviewer 4 Report

I found the paper to be interesting for researchers in the hydro-meteorological area.

Author Response

Realizada uma revisão ortográfica no idioma inglês. E feitas as alterações indicadas abaixo no texto.

Page 3 - line 94 - Table 1: completed and updated for English

Page 5 – 185 lines – changed the equation

Page 9 - line 276 is OK

Page 10 - line 281 included (B)

Page 11 – lines 357 and 358: added text

Page 14 – line 405 – landslide events and their scars were not the objects of this article

Page 14 – lines 441 to 453 – added text

Round 2

Reviewer 1 Report

Although the article's main content has remained the same, the author has made minor improvements to the manuscript.

The only added value of this manuscript is the recurrence frequency of extreme rain. Discussions related to the physical basis of variation of recurrence frequency must be further strengthened. Currently, the results chapter is minimal, and the discussion chapter still needs improvement.

For now, the manuscript is not suitable for publication. Discussions related to the physical basis of the recurrence frequency variation must be strengthened, related to climate change and other physical processes. In addition, the introductory chapter is still poor and inadequate as a scientific journal. Please strengthen by clarifying the research gap. The conclusions are also poor; their contents are mainly discussions that are not derived from the results of the research that you did.

Author Response

- Introduction, line 25

- Formulas (3) and (4), letter increase (font)

Reviewer 2 Report

The manuscript has improved a lot and now it can be recommended for publication.

Author Response

(The authors gave the same response as above.)
